

# Can global precipitation datasets benefit the estimation of the area to be cropped in irrigated agriculture?

Alexander Kaune[1,2], Micha Werner[1,3], Patricia López López[3,4], Erasmo Rodríguez[5], Poolad Karimi[1], Charlotte de Fraiture[1,2]

[1]Water Science and Engineering Department, IHE Delft Institute for Water Education, Delft, the Netherlands
[2]Wageningen Institute for Environment and Climate Research, Wageningen University & Research, the Netherlands
[3]Division of Inland Water Systems, Deltares, Delft, the Netherlands
[4]Division Department of Physical Geography, Faculty of Geosciences, Utrecht University, Utrecht, the Netherland
[5]Grupo de Investigación en Ingeniería de Recursos Hídricos (GIREH), Universidad Nacional de Colombia, Bogotá, Colombia

*Correspondence to*: Alexander Kaune (alex.kaune83@gmail.com)

**Abstract.** The area to be cropped in irrigation districts needs to be planned according to the available water resources to avoid agricultural production loss. However, the period of record of local hydro-meteorological data may be short, leading to an incomplete understanding of climate variability and consequent uncertainty in estimating surface water availability for irrigation area planning. In this study we assess the benefit of using global precipitation datasets to improve surface water availability estimates. A reference area that can be irrigated is established using a complete record of thirty years of observed river discharge data. Areas are then determined using simulated river discharges from six local hydrological models forced with in-situ and global precipitation datasets (CHIRPS and MSWEP), each calibrated independently with a sample of five years extracted from the full thirty year record. The utility of establishing the irrigated area based on simulated river discharge simulations is compared against the reference area through a pooled Relative Utility Value. Results show that for all river discharge simulations the benefit of choosing the irrigated area based on the thirty years simulated data is higher compared to using only five years observed discharge data, as the statistical spread of *PRUV* using thirty years is smaller. Hence, it is more beneficial to calibrate a hydrological model using five years of observed river discharge and then extending it with global precipitation data of thirty years as this weighs up against the model uncertainty of the model calibration.

## 1 Introduction

As water becomes scarce, efficient decision making based on solid information becomes increasingly important (Svendsen, 2005). Solid information on climate variability and climate change are key to adequately estimate the availability of water for human livelihoods, the environment and agricultural development (Kirby et al., 2014, 2015), especially for irrigated agriculture, which by volume is the largest user of freshwater (de Fraiture and Wichelns, 2010). Available climatological records used for estimation of water resources availability in the irrigation sector are, however, often short (Kaune et al., 2017), and may not be representative of the full distribution of climate variability. This may particularly be so in developing countries, where the need to develop irrigation areas is the greatest, and can lead to sub-optimal decisions, such as the overestimating or





underestimating of the area that can be planted. Local authorities deciding on the irrigated area clearly prefer to use the true record of climate variability to estimate the adequate irrigation area to be able to justify their decision based on expected economic benefits, but these records may often be short.

Recent studies show that hydrological information from remote sensing datasets can be effectively used for estimation of
surface water availability (Peña-Arancibia et al., 2016), water accounting (Karimi et al., 2013) and to help improve detection of droughts at basin scale (Linés et al., 2017). Combined with local data these datasets can potentially provide improved information to support decisions in irrigated agriculture. Global hydrological models have been used to estimate the river discharge at basin level for the development of irrigated areas and assess the risk of water scarcity (Kaune et al., 2018), and although these show promising results in large basins, the use of a calibrated local hydrological model may be more suitable
in smaller basins (López López et al., 2016) as a finer spatial resolution may then be used and local hydrological processes better represented.

Such local models will typically require some level of calibration, and the challenge is to calibrate these when the period of record of the observed data from available in-situ stations is limited. If the period of record is short, then the data may not provide full representation of the true climatic variability, and the water resource estimate will be conditional on whether the
available data is from a relatively wet, normal or relatively dry period. This is particularly relevant in climates that are influenced by phenomena such as the El-Niño Southern Oscillation (ENSO).

Using hydrological models forced by a longer period of record from available precipitation datasets may help improve discharge estimates for reliably determining the irrigated area, as the climatic variability can be better represented. However, model uncertainty, as well as the uncertainty of the representativeness of the model given the data used in model calibration
will need to be taken into account. Recently, several global precipitation datasets have become available, based on remote sensing as well as re-analysis models, with periods of record spanning thirty plus years. Examples include the CHIRPS precipitation dataset (Funk et al., 2015), which integrates in-situ meteorological data and global earth observations, and the recently developed MSWEP precipitation dataset (Beck et al., 2017b), which integrates in-situ meteorological data, global earth observations and the ERA-Interim re-analysis datasets. Both have been widely used to assess water availability and the
risk of water scarcity and drought events (López López et al., 2017; Shukla et al., 2014; Toté et al., 2015; Veldkamp et al., 2015).

Despite the opportunities these modern datasets offer, they have largely been neglected by the irrigation sector for the estimation of water resources availability and variability (Turral et al., 2010), which relies primarily on in-situ datasets, even when the availability of these datasets is often limited. Assessing the potential benefit of combining data from available in-situ
stations, global earth observations and reanalysis datasets to better estimate surface water availability can therefore be of considerable value to irrigation managers.

In this paper we hypothesise that the simulated river discharge for a period of record of thirty years using a calibrated local model forced by datasets such as CHIRPS or MSWEP provides more reliable estimates of water resources availability and the area to be irrigated, than when considering the shorter time series of observed discharge that is used to calibrate the model.



This is evaluated through an extended version of the hydro-economic Expected Annual Utility framework that determines the value of using each of the different datasets in determining the areas that can be irrigated as a function of the estimated availability of water.

## 2 Methods

### 2.1 Coello Irrigation District, Colombia

We apply our analysis to the Coello Irrigation District in Colombia. The Coello Irrigation District is an existing irrigation district located in the upper Magdalena basin, in the Tolima Department, a region subject to considerable climate variability and that is vulnerable to droughts (IDEAM, 2015). The irrigation district serves an irrigated flatland of approximately 250 km², comprising mainly of irrigated rice, which is sowed continually throughout the year (Urrutia-Cobo, 2006). The water available for irrigation depends on the total discharge of two rivers from neighbouring mountainous basins; the Coello and Cucuana Rivers (Figure 1). The Coello basin has an area of 2000 km², and the Coello River has a length of some 112 km starting at 5300 m.a.s.l. and flowing into the Magdalena River at 280 m.a.s.l. with an average flow of 23 m³/s (Vermillion and Garcés-Restrepo 1996). The Cucuana basin has similar physical characteristics to the Coello basin. In this research, we focus only on the Coello basin to estimate the surface water availability for irrigation, as the available discharge data from the Cucuana River (Corea Station) is too short for the purpose of our experiment.

### 2.2 Hydro-meteorological data

In-situ precipitation and temperature data were obtained from the network of meteorological stations operated by the *Instituto de Hidrología, Meteorología y Estudios Ambientales (IDEAM)*, the Colombian hydro-meteorological institute, and interpolated to a gridded dataset with 0.1° spatial and daily temporal resolution for the whole Magdalena-Cauca basin (Rodriguez et al., 2017). The temperature data was used to estimate potential evapotranspiration with the Hargreaves method (Hargreaves, 1994).

Two global precipitation datasets were considered: (i) the Climate Hazards Group InfraRed Precipitation with Station data (CHIRPS, Funk et al., 2015) and (ii) the Multi-Source Weighted-Ensemble Precipitation (MSWEP; Beck et al., 2017). CHIRPS precipitation is a remotely sensed and ground corrected dataset available globally at 0.05° resolution, while MSWEP precipitation is a merged gauge, satellite and reanalysis dataset available globally at 0.25° resolution. All precipitation, temperature and potential evapotranspiration datasets are available for the 1983-2012 period.

Daily river discharge data for the 1983-2012 period was obtained from the stations operated by IDEAM at the Payande station (21217070) in the Coello River (Figure 1).



## 2.3 Hydrological modelling

The Dynamic Water Balance Model (Zhang et al., 2008), a lumped conceptual hydrological model based on the Budyko framework (Budyko, 1974), was selected to simulate the river discharge in the Coello basin at monthly time scale. The Dynamic Water Balance Model has been applied in several basins around the world (Kaune et al., 2015; Kirby et al., 2014;

Tekleab et al., 2011; Zhang et al., 2008), showing reliable river discharge simulations at monthly time scale. The model has a simple structure without routing, simulating the basin hydrological processes with a reduced number of parameters. There are only four model parameters; basin rainfall retention efficiency $\alpha_1$ (-), evapotranspiration efficiency $\alpha_2$ (-), recession constant $d$ (1/month); and maximum soil moisture storage capacity $S_{max}$ (mm). Low (high) values of basin rainfall retention efficiency or evapotranspiration efficiency implies more (less) direct runoff. The recession constant $d$ characterises baseflow, with

parameter values ranging between zero and one. The maximum soil moisture storage capacity relates to the root soil depth and soil texture of the basin. As the Coello basin is small, routing processes can be ignored when estimating monthly water availability, which is calculated as the accumulated runoff in the basin upstream of the point of interest.

In this study, surface water availability for irrigation was established as the discharge in the Coello River, considering an environmental flow of 25% from the available water resources. A maximum soil moisture storage capacity $S_{max}$ of 176 mm

was determined for the Coello basin based on physical soil and vegetation characteristics. The hydrological model was forced with the different precipitation datasets (described in detail in section 2.2). Although river flow data was available for the full 1982 through 2012 period, to explore the influence of limited availability of observed discharge data, six independent samples of five years were extracted from the thirty year dataset (1983-1987, 1988-1992, 1993-1997, 1998-2002, 2003-2007 and 2008-2012) for calibration of the model parameters (Figure 2). These samples were extracted as contiguous samples of five years to

represent different climatological periods, and were applied to calibrate six sets of models, each using one of the observed discharge samples. 10000 model parameter sets ($\alpha_1$, $\alpha_2$, and $d$ with values uniformly distributed between 0 and 1) were generated, and subsequently forced with the full dataset of thirty years of in-situ precipitation data (1983-2012). From this Monte Carlo simulation, the five best performing model parameter sets are selected for each of the five year samples based on the comparison of the simulated and observed discharges for the corresponding period. Model performance is measured using

the Kling-Gupta Efficiency (KGE) metric (Gupta et al., 2009). This resulted in thirty calibrated models, which were used to provide simulated discharge data at the Payande Station for the full thirty year period, forced by each of the three precipitation datasets. Performance metrics of the mean discharge simulation of the five models were calculated separately for each month across the six periods to evaluate the hydrological performance, including KGE, Pearson's correlation coefficient (r), and percent bias (Pbias).





## 2.4 Determining the irrigated area

Similar to Kaune et al. (2018), the area that can be irrigated is determined based on an operational target monthly water supply reliability ($R$ =75%), which means that the monthly demand is met for on average 75% of the years (Eq. 1). The monthly irrigation demand varies depending on the irrigation area, which is the variable to be obtained. A fixed demand rate of 0.2

m³/s/km² was used in the Coello Irrigation District (Kaune et al., 2017).

$$R \leq p\{(Q_a - A_i \cdot r) \geq 0\} \tag{1}$$

where R is the water supply reliability (or probability of non-occurrence of water scarcity, $pr\{NWS\}$), $p$ is the relative frequency, $Q_a$ is the multi-annual monthly surface water availability (considering an environmental flow of 25% from the available water resources), $A_i$ is the planned irrigation area for dataset $i$ (simulated or observed), and $r$ is the demand rate.

The water availability distribution for each calendar month is established using the multi-annual monthly river discharge, which may be obtained either from the observed or the simulated data. Given the small sample size of 30 years, the empirical distribution of water availability is obtained by applying a bootstrap resampling with replacement procedure, with the size of the bootstrap set at 25,000. The size of the bootstrap is determined iteratively using a progressively increasing sample size until a stable estimate of the empirical distribution is achieved. A reference irrigated area is established using the empirical

distribution derived from the observed monthly river discharges of thirty years (1983-2012). The areas that can be irrigated for each of the six calibrated models is similarly determined but now using the discharge simulations for the full thirty year period. Irrigated areas are additionally obtained for the six five-year samples of observed discharge, and for comparison also using the five year period of simulated discharges for each of the six calibrated model, where the period is commensurate with the period used for calibration. For each irrigation area that is obtained, the real probability of water scarcity is determined

using the observed surface water availability (which is also a multi-annual monthly bootstrap resample), and the demand calculated using the estimated area (Eq. 2).

$$pr\{WS\} = p\{(\widehat{Q_a} - A_i \cdot r) < 0\} \tag{2}$$

where $pr\{WS\}$ is the probability of occurrence of water scarcity, $p$ is the relative frequency, $\widehat{Q_a}$ is the observed surface water availability, $A_i$ is the planned irrigation area obtained from Eq. 1, and $r$ is the demand rate. These probabilities are then used

to determine the expected annual utility to evaluate the economic value (Table 1 and section 2.5).

## 2.5 Evaluating the cost of choosing the irrigation area

The cost of choosing the irrigation area was evaluated with an extended version of the hydro-economic framework developed by Kaune et al. (2018)based on the economic utility theory (Neumann and Morgenstern, 1966). The cost is calculated as the opportunity cost when the irrigation area is selected to be too small, or the production loss due to water scarcity when the

irrigation areas is selected to be too large. When the area selected is equal to the reference area, then the cost is zero. Similar





to Kaune et al. (2018), the Relative Utility Value, $RUV$ is used to compare the expected annual utility between the reference and the irrigated area derived using either the simulated discharge or the shorter five year observed discharge sample (Eq. 3).

$$RUV = \frac{(U_i - U_r)}{U_r} \tag{3}$$

where $U_r$ is the expected annual utility (revenue) obtained with the reference irrigation area from observed river discharge, and $U_i$ is the expected annual utility obtained with any of the irrigation areas obtained from the discharge simulations described in section 2.4.

The expected annual utility $U$ is defined as the expected annual crop production, given monthly probabilities of (non-) water scarcity, and considering a loss in crop production if water scarcity does happen in any one month (Eq. 4).

$$U = pr\{WS\} \cdot (P_{NWS} - L_{WS}) + (1 - pr\{WS\}) \cdot P_{NWS} \tag{4}$$

where $pr\{WS\}$ is the monthly probability of water scarcity defined in section 2.4; $P_{NWS}$ is the expected annual crop production ($P_{NWS} = c \cdot A_i \cdot y_e$) which includes the irrigation area $A_i$ obtained from section 2.4 and converted into hectares, price of the crop per ton ($/t) and the expected crop yield $y_e$ (t/ha); and $L_{WS}$ is the annual production loss if water scarcity happens in any one month.

For determining the annual production loss $L_{WS}$ an approach is applied where each month corresponds to the growth stage distribution of the crop based on information provided by the Coello Irrigation District. We assume that only rice is grown in the Coello Irrigation District with a growing length of four months sowed over the entire year. The loss in annual rice production $L_{WS}$ due to water scarcity happening in any one month is determined with Eq. 5:

$$L_{WS} = c \cdot A_i \cdot \sum_{month}(y_e - y_a) \tag{5}$$

where $y_e$ is the expected harvested crop yield in a month (t/ha) and $y_a$ is the actual harvested crop yield in a month (t/ha) due to water shortage happening in any one month. Water shortage happening in any one month of the four month crop period will lead to a yield reduction. The actual harvested crop yield $y_a$ obtained, is determined with the FAO water production function in Eq. 6 (FAO, 2012):

$$\left(1 - \frac{y_a}{y_e}\right) = K_y \left(1 - \frac{ET_a}{ET_p}\right) \tag{6}$$

where $K_y$ is the weighted average yield reduction value per month calculated from established yield reduction factors due to water deficit for each growth stage of rice (FAO, 2012), and the distribution of growth stages as reported by the Coello Irrigation District; $ET_a$ is the actual evapotranspiration and $ET_p$ is the potential evapotranspiration. In our experiment, the actual evaporation is unknown, as this will depend on irrigation scheduling and practice as well as precipitation. As this detail is beyond the scope of this paper, we assume the reduction in evapotranspiration $\left(1 - \frac{ET_a}{ET_p}\right)$ to be 20% for the reference irrigation area when water shortage occurs. 20% is selected as this is the evapotranspiration deficit that rice farmers can easily





cope with (FAO, 2012). To account for the increased deficit for irrigation areas selected to be larger than the reference area, the evapotranspiration reduction is increased proportionally, assuming the available water is uniformly distributed in the new irrigation area. For irrigation areas selected to be smaller the reduction is decreased proportionally.

An average price of 329 $/t and an expected rice yield of 6.8 t/ha based on national statistics (DANE, 2016; Fedearroz, 2017)
are used to estimate the expected annual rice production.

If $RUV$ is equal to zero, then the expected annual utility obtained with the reference and simulated irrigation areas are the same, and there is thus no cost associated to using the simulated information. A negative $RUV$ entails an opportunity cost due to the planning of too small an irrigation area (defined as cost type 1). A positive $RUV$ entails an agricultural loss due to the area being planned larger than can be supported by water availability and water shortages thus occurring more frequently than
expected (defined as cost type 2). The statistical spread of $RUV$ is derived from the bootstrap resample. The spread depends on the probability of water shortage being larger compared to the reference and on the yield response factor, entailing that the production loss incurred depends not only on the increased occurrence of water shortage, but also on the sensitivity of the crop to water deficit.

$RUVs$ are pooled as to give a Pooled Relative Utility Value ($PRUV$) to evaluate the cost of choosing the irrigation area from
the six possible irrigation areas obtained for a river discharge simulations. This is done as it is not *a-priori* clear when only five years of observed data are available, from which part of the full climatological record these may be. The Pooled Relative Utility Value ($PRUV$) is a concatenated vector of the Relative Utility Value obtained for each calibration sample (Eq. 7).

$$PRUV = \{ RUV_1 \parallel RUV_2 \parallel \cdots \parallel RUV_6 \} \tag{7}$$

where $PRUV$ is the Pooled Relative Utility Value and $RUV_x$ is the Relative Utility Value for each calibration sample (in this
case six samples).

Similar to $RUV$, the $PRUV$ is a hydro-economic indicator that can be larger (cost type 2), equal (no cost) or smaller than zero (cost type 1). The statistical spread of $PRUV$ encompasses the variability of $RUV$ among the six calibration samples. If the statistical spread of $PRUV$ is large, then the variability of planned irrigation areas is large among samples. This means that the cost of choosing the irrigation area based on the available information is high. If on the other hand the statistical spread of
$PRUV$ is small, then the variability of planned irrigation areas is also small and the cost of choosing the irrigation area based on the available information is low.

## 3 Results

### 3.1 Discharge simulations

The monthly observed and the simulated discharges calculated with the different precipitation datasets from the calibration
samples are shown in Figure 3, Figure 4 and Figure 5 (only the samples for the 1993-1997 and 1998-2002 periods are shown), and in the supplementary material (all samples). Discharge simulations change depending on which precipitation dataset is



used as forcing and which sample is used to calibrate the hydrological model. In general, however, the mean discharge simulations show an overall agreement with observations.

The performance metrics for each month are shown in Figure 6 and in the supplementary material. In all months using the discharge simulations with different precipitation datasets, positive KGE values are obtained with the exception of simulations

with MSWEP in April and November, which are both wet season months. In February (dry season) the highest KGE value is obtained using the simulations with observed precipitation (0.75). For all samples in February (dry season), the KGE value is higher for discharge simulations with observed precipitation and CHIRPS than those using MSWEP, with exception of one sample (2008-2012).

In terms of Pbias, simulations with MSWEP consistently overestimate the discharge between October and May for all samples.

The largest overestimation occurs in April (wet season) (Pbias=75%). For simulations with observed precipitation and CHIRPS, monthly Pbias follows a similar trend, overestimating discharge in April for most samples and underestimating discharge between January and April for only two samples (1983-1987 and 1998-2002). Between May and September underestimated discharges are obtained using simulations with In-Situ and CHIRPS for all samples. Simulations with MSWEP in June and July are also underestimated with exception of sample 1998-2002 (Pbias positive for all months).

The correlation values vary among simulations and for each month. The correlation values range between 0.25 and 0.85. In February, using In-Situ precipitation, correlation values are above 0.6. Simulations with CHIRPS and MSWEP results in correlation values between 0.7 and 0.8 in February. The largest difference between correlations occurs in March (CHIRPS correlation is 0.5, MSWEP correlation is 0.6, and In-Situ correlation is 0.8).

Simulations with In-Situ precipitation and CHIRPS are found to be behave similarly, which is not surprising as CHIRPS uses

station corrected data. MSWEP also includes station corrected data, but it is derived in part from the ERA-Interim data which in itself is not good at capturing convective precipitation (Leeuw et al., 2015). This explains the poor simulation performance with MSWEP in April and November as these are wet months in a tropical region with predominant convective precipitation. As our work is focused on determining the critical irrigation area under monthly water scarcity, we are less concerned with the simulation performance in wet months, but focus rather on the more critical dry months (e.g. February), which have shown to

perform well for the selected precipitation datasets.

## 3.2 Estimating the area that can be irrigated

The areas that can be irrigated based on the water availability of the Coello river are established using the simulated discharges from section 3.1., a defined environmental flow, a fixed demand rate per unit area cropped, and a water supply reliability target of 75%. Irrigation areas are established for the reference discharge (observed thirty years); for each of the thirty year discharge

simulations using the models derived with each calibration sample; as well as using the observed discharges for each of the six five-year samples. Finally, for comparison, irrigated areas are derived using only five years of simulated data for each of the six five-year samples, where the simulated five years are the same as the five years used in calibration. The areas that can be irrigated given the simulated (or observed) discharges are found to vary significantly when compared to the reference



irrigated area (which was established as 67.45 km²), with areas ranging from 2% to 40% smaller, to 1% to 69% larger (Table 2). In the case of the thirty year simulations, the irrigation areas obtained are found to be always larger than the reference area, with the overestimation ranging from 3% to 69% (Table 2), with exception of one sample where an underestimation of 3% when using the observed precipitation is found. The largest estimates of areas that can be irrigated is obtained with MSWEP

simulation (69%), which agrees with discharge model performance (section 3.1), as KGE and r values are the lowest for this model, while the bias values are the highest of all the simulations. For simulations with CHIRPS and observed precipitation, Pbias is positive for sample 1998-2002 in the dry months (e.g. Februrary), leading to an overestimation of the irrigation area of 40%. For sample 1993-1997, Pbias is negative for these simulations (close to -10% in February), resulting in a lower estimation of the irrigated area. In this case, an underestimation of 3% in the area is found for the simulation with observed

precipitation, but an overestimation of 3% in the area is found for the simulation with CHIRPS. This is related to the difference in variability as the water availability is derived based on the distribution and not on the mean.

The areas that can be irrigated that are obtained using the observed discharges for each of the six five year periods show relatively small variation when compared to the reference area, ranging from 19% smaller to 11% larger. The average area of the six five year samples is slightly smaller at 64.99 km², just 2.5% smaller than the reference. Conversely, the areas derived

using the simulations for each of the five year periods shows that these vary quite considerably, with an overestimation ranging from 10% to 62% and underestimation ranging from 9% to 40% across all precipitation sources. This range is comparable for all three precipitation forcing datasets, indicating that the variability can be attributed primarily to model error, conditional on the five year dataset used in calibration.

### 3.3 Probability of water scarcity

The probability of water scarcity using the irrigation areas obtained for each of the simulated and observed discharges for the five year periods as well as for the reference are shown in Figure 7, Figure 8, Figure 9 and Figure 10 (samples 1993-1997 and 1998-2002) and in the supplementary material (all samples). The probability of water scarcity using the irrigation area obtained using the observed discharges are shown in Figure 7. As expected, the probability of water scarcity in February, which is the most critical month, shows a median value equal to 25% and probabilities lower than 25% for the other months when using

the irrigation area obtained with the reference discharge (30 years). The spread of the probability of water scarcity indicated by the box-whiskers plot, showing the mean, interquartile range and minimum and maximum, is due to the distribution of the bootstrap, representing the uncertainty in the estimate due to the 30 year period of record.

Figure 8 similarly shows the probability of water scarcity for irrigation areas obtained using observed discharges for the five year periods; 1993-1997 and 1998-2002 (results for the other four periods included in the supplementary material). This shows

that for the period 1993-1997, the median value is lower than 25% for all months, while for the period 1998-2002 the median value is higher than 25% for January and February. This reflects the smaller, respectively larger irrigated areas established with each of these datasets. Figure 9 shows the probability of water scarcity for irrigation areas obtained using the discharge simulations of thirty years. The probability of water scarcity in February shows median values higher than 25%, commensurate




with the overestimation found in the hydrological model, with the exception of one simulation using observed precipitation, calibrated with the 1993-1997 sample of observed discharge data. Between April and June and in October/November, using the irrigation areas obtained with the discharge simulations, the probability of water scarcity is always found to be lower than 25%, as these are the two wet seasons of the bimodal climate. For all samples, the probability of water scarcity is the highest

for the simulations using MSWEP precipitation. Using the irrigation areas obtained from the simulations calibrated with the 1983-1987 and 1998-2002 samples show higher probabilities of water scarcity for all months when compared to the simulations calibrated with the other samples. This reflects that these years were relatively wet, influencing discharge simulations and resulting in larger irrigation areas being selected. The pattern for sample 1993-1997 is more similar to the pattern found using the reference area found with the 30 years observed discharge.

The probability of water scarcity for irrigated areas obtained with simulated discharges of only five years are shown in Figure 10 (again results for the 1993-1997 and 1998-2002 samples are shown, with the remaining four periods provided in the supplementary material). The monthly probabilities of water scarcity show large differences between samples. In this case, four out of the six samples do not show median probability of water scarcity higher than 25% for any month, meaning that the irrigation area is underestimated compared to the reference. For the 1998-2002 sample, the probability of water scarcity is the

highest with a median probability of water scarcity between 50% and 75% in February.

### 3.4 Relative Utility Value

The annual expected utility is calculated using the economic return of the rice crop and the estimated yield determined using the irrigated areas established with the simulated discharge information, and the probability of water scarcity in each month for the 30-year period based on the observed discharges. Relative Utility Values are then found through comparing these

against the annual expected utility calculated using the reference area and discharge information.

Figure 11 shows the Relative Utility Values obtained for areas determined using the five year samples of observed discharge for the 1993-1997 and 1998-2002 periods (again the remaining four periods are provided in the supplementary material), with median estimates of -0.02 and 0.11, respectively.

Relative Utility Values obtained for areas determined using discharge simulations of thirty years are shown in Figure 12 (and

in the supplementary material), with median estimates between -0.03 and 0.65. The Relative Utility Values closest to zero are found for simulations using both the observed and the CHIRPS precipitation datasets, -0.03 and 0.03, respectively, both when using the 1993-1997 sample for model calibration. Of the six samples, this five year period was already noted to be most representative of the whole 30-year period.

For all samples, the Relative Utility Values for simulations using the MSWEP dataset are found to be largest, with values

between 0.3 and 0.65, indicating a higher production loss due to the higher probability of water scarcity. For simulations using the 30-year observed precipitation, consistent median values between 0.18 and 0.45 are obtained, with the exception of one sample (-0.03). Those obtained with CHIRPS simulations are consistent with those found using the observed precipitation.





The Relative Utility Values obtained using irrigated areas determined with the simulated discharges of only five years (Figure 13 and supplementary material), show median estimates between -0.2 and 0.6 (MSWEP), which are larger than the simulations of thirty years. The *RUV* closest to zero are found for simulations with CHIRPS (-0.09), while results for simulations using the observed precipitation show more consistent values closer to zero. In this case, results show more negative *RUVs* for each

simulation forcing, and results are less consistent between samples compared to the results obtained with the thirty years.

For the 1993-1997 period, the *RUV* obtained for the irrigated area determined with observed discharge of five years (-0.02) and the *RUV* obtained with for the area determined with simulated discharges of 30 years using either the CHIRPS or the observed precipitation (0.03 and -0.03) are similar and close zero. The extended precipitation period compensates the model uncertainty and results in reliable *RUV* estimates.

A large statistical spread in *RUV* is found in months where the probability of water scarcity is higher than the reference. This is clearly shown for MSWEP simulations, which have the largest estimates of the irrigated area. For months where the probability of water scarcity is lower than the reference, the statistical spread in *RUV* is low. In these cases the statistical spread of *RUV* is a result only of the spread of the reference annual expected utility, resulting from the distribution of the probability of water scarcity. The statistical spread of the *RUV* values is lower when the simulated annual expected utility and the reference

annual expected utility are more similar, which means that the *RUV* is closer to zero as shown when using the 1993-1997 sample. An absence of statistical spread for the *RUV* values reflects zero probability of water scarcity in both the simulated and the reference expected annual utility.

Even though the probability of water scarcity is not the highest in November, the statistical spread of the *RUV* is the largest when water scarcity happens in that month. This is due to the high sensitivity of the crop to water deficit ($K_y$=1.4) in November,

which then becomes the determining factor for obtaining a large statistical spread. On the other hand, the smallest statistical spread, or no statistical spread of the *RUV* is found when water scarcity happens in February or May. We select February, May and November as the representative months for further analysis with the Pooled Relative Utility Value.

### 3.5 Pooled Relative Utility Value

The Pooled Relative Utility Value *PRUV* is obtained from the *RUV* values for each of the six samples in section 3.4. In Figure

14 and Figure 15, the *PRUV* results for areas estimated using the observed discharges, and for the simulated discharges for five and thirty years are shown for November, February and May. These are the representative months identified in section 3.4, with similar results found for *PRUVs* when water scarcity happens independently in each month.

The statistical spread of *PRUV* represents the risk of randomly choosing one irrigation area out of the six possible irrigation areas given by the six calibration samples of five years. Results for the five year simulations show a large statistical spread of

the *PRUV* values, with the distribution positively skewed. This skewness is due to the influence of one high RUV sample out of the six RUV samples, resulting in a maximum positive *PRUV* value for each precipitation dataset 0.18, 0.25, and 0.6. The statistical spread of *PRUV* for the five year simulations with MSWEP precipitation is the largest among the simulations,



implying that the cost of choosing the irrigation area using this dataset is the highest. Using thirty years of simulated discharges does reduce the statistical spread in *PRUV* when compared to the five year simulations. For observed precipitation, simulations with five years show the range of *PRUV* to be between -0.38 and 0.18, with an interquartile range between -0.3 and -0.15; while simulations with thirty years show a the range of *PRUV* to be between -0.03 and 0.5, with an interquartile range between

0.18 and 0.4. For CHIRPS and MSWEP precipitation, the reduction in the statistical spread in *PRUV* with thirty year simulations is more evident. For the thirty year simulations, the smallest statistical spread in *PRUVs* is found for CHIRPS precipitation (between 0.03 and 0.4). This means that the cost of choosing the irrigation area is lower when using simulations with CHIRPS compared to simulations with In-Situ and MSWEP precipitation. In addition, using CHIRPS leads to median *PRUV* values closer to zero, thus choosing among the irrigation area samples results in an irrigation area closer to reference

irrigation area.

The statistical spread in *PRUV* when using observed discharge of five years (-0.20 to 0.12) is similar to the statistical spread in *PRUV* when using the best simulation with thirty years (CHIRPS, 0.03 to 0.4). Again using the longer precipitation record to provide a longer record of simulated discharge results in a reduction in the statistical spread in *PRUV* and compensates the model uncertainty. This means that using the thirty year simulation is beneficial, as the cost of choosing the irrigation area is

similar to the cost when using the five year observed discharge.

**4 Discussion**

The Pooled Relative Utility Value *PRUV* used in this study is defined as a joined vector of six samples of the Relative Utility Value. This value includes the irrigation areas for river discharge simulations derived using different precipitation datasets, the monthly probability of water scarcity using these areas, and the potential yield reduction due to water deficit for rice.

Results of *PRUV* show that using the CHIRPS global precipitation dataset in discharge simulations reduces the risk of choosing the irrigation area compared to discharge simulations with In-Situ and MSWEP precipitation.

In the Coello basin we have the good fortune to have a long period of record of hydrological data (1983-2012) to use as reference for establishing the climatological availability and variability of the available water resource. This may not be the case in other basins. Water resources estimation may then need to be done with the limited information that is available. To

help understand the risk of estimating the available water resources when only limited information is available, we used observed discharge with a shorter period of record (five years) to calibrate a local hydrological model and apply this to obtain simulated discharge with a longer period of record (30 years) either using a precipitation dataset based on observed data (Rodriguez et al., 2017) or a global precipitation dataset, including CHIRPS and MSWEP (Beck et al., 2017; Funk et al., 2015;). We establish six samples of five years to calibrate the parameters of a hydrological model, and simulate six possible

discharges of thirty years to imitate a setting where information about how representative the short record of available observed discharge is not known *a-priori*. For each sample the annual expected utility is determined including the monthly probability of (non-) water scarcity using different irrigation areas from different discharge simulations and the annual crop production




with water scarcity (not-) happening in a month. Positive and negative Relative Utility Values were found with different discharge simulations. Positive values indicate a crop production loss due to unexpected water scarcity for too large an irrigation area being planned. Negative values indicates an opportunity cost due to the planning of too small an irrigation area. Results show that the *RUV* varies depending on in which month water scarcity happens. While the spread in the estimates of

probability of water scarcity is found to be the largest in the month of February, the spread in *RUV* is larger when water scarcity happens in November. This is due to the difference in sensitivity of the crop yield to water deficit, depending on the growing stage of the crop. In the Coello basin, rice has an average growing length of four months and is sowed during the entire year. This means that if water scarcity does happen in a particular month, four different growth stages will be affected each with a different yield reduction factor resulting in an average yield reduction value. If water scarcity happens in November the average

yield reduction value from the four growing stages of rice is 1.4. This means that the average yield reduction in November under an equal degree of water deficit is 1.75 times higher than in February ($K_y$=0.8). If water scarcity occurs in February, even though the probability of water scarcity is higher than the reference 25%, the statistical spread in *RUV* is low due to the low average yield reduction value. Using different sources of river discharge information to estimate the irrigation area will indeed change the estimates of the monthly probability of water scarcity changing the *RUV* values. However, the impact on

annual production may be low if water scarcity occurs in the month where the sensitivity of the crop to water deficit is low. Reducing agricultural production losses depends not only on using adequate river discharge information to estimate the irrigation area, but also on adequate planning of the crop stage distribution.

For an irrigated area selected based on the estimate of water availability using simulated discharges, a decision maker takes an additional risk due to not knowing *a-priori* how representative the data used for calibrating the model is of climatic variability.

This is why we introduce the Pooled Relative Utility Value *PRUV* in order to evaluate the risk of choosing an irrigation area derived from different river discharge simulations. If the statistical spread of *PRUV* is low (high), then the cost incurred by choosing an irrigated area based on the results of the simulations is equally low (high). The Pooled Relative Utility Value results using the global precipitation CHIRPS showed a lower cost in choosing the irrigation area compared to *PRUV* results using both a dataset based on observed precipitation, as well as the MSWEP global precipitation dataset. This would suggest

that the CHIRPS precipitation should be used instead of both observed and MSWEP precipitation when determining the surface water availability for irrigation area planning to avoid the risk of agricultural production loss due to a poorly chosen irrigated area that can be supported based on water availability. This is not a general conclusion, as it is closely related to how representative the precipitation dataset used is of the true precipitation amount and variability in the basin. The CHIRPS dataset does include observed data (Funk et al., 2015), which is the same as that used in our study to establish the In-Situ precipitation

dataset. In that sense, it is also an interpolated dataset, but with additional information from the satellite. This may well provide additional detail on the variability of precipitation in a tropical mountainous basin such as the Coello.

Interestingly, the performance of the model using the observed precipitation dataset is similar to that of the model using the CHIRPS precipitation dataset when considering common model performance statistics such as Kling-Gupta Efficiency (KGE), percentage bias (Pbias) and the correlation coefficient (r). The MSWEP product includes reanalysis datasets additional to





observed and satellite datasets, but instead of providing a benefit its local application in this small to medium sized basin in Colombia has a negative influence on the representation of the climate variability. In that sense, our results match with previous research where the performance of reanalysis datasets in regions dominated by tropical warm rain processes is not the best (Beck et al., 2017b) attributed in part to the poor prediction of convective precipitation (Leeuw et al., 2015). Moreover, the

0.25° resolution might be too coarse to represent the spatial variability in the basin which undervalues the potential use of these datasets in such conditions. The further development of the MSWEP dataset, including an improved resolution of 0.1 degrees may increase its value for applications such as that explored in this paper (Beck et al., 2017a). Additionally, the period of record of consistent data for datasets such as MSWEP, but also of CHIRPS continues to increase. For now, in Colombia, where the availability of observed precipitation is reasonable (IDEAM, 2015; Kaune et al., 2017), the CHIRPS datasets appears

provide the best estimates of surface water availability in basins larger than 2000 km² for determining the irrigation area. The benefit of CHIRPS, MSWEP and other such global precipitation dataset can be evaluated in other case studies around the world using the proposed framework. Certainly there is a new opportunity for the irrigation sector in using modern hydro-meteorological data and information to improve water allocation decisions considering the economic impacts of uncertainty in those datasets.

An irrigation manager may be reluctant to use simulated information instead of the observed until it is proven that the additional period of record of precipitation from for example global datasets compensates the uncertainty of the use of a hydrological model. In that sense, *PRUV* results provide evidence that using discharge simulations with thirty year precipitation (CHIRPS) is equivalent in using observed discharge of five years as the risk of choosing the irrigation area is similar. As the period record of datasets such as CHIRPS increases, this risk will be expected to reduce further.

Using a longer period of record of observed discharges will help make better estimates of the irrigated area that can be supported by the available water resources, but when the availability or quality of observed discharge is limited, extending the period of record using model based discharge simulations provide an alternative to estimate the area to be cropped. The results of the model used in the Coello basin also show that the overestimation or underestimation of the planned irrigation area depends in part on the model bias, particularly in the ability of the calibrated model to provide reliable simulations for low

flow periods, which are the most critical in this application. In the case presented here, we use a very simple model structure, and using a simulated discharges from an enhanced model structure can be explored to obtain more accurate results.

## 4 Conclusion

We apply an extended hydro-economic framework to assess the benefit of using global precipitation datasets in surface water availability estimates to reduce the risk of choosing the area that can be irrigated with available water resources based on

limited available information. We estimate irrigation areas using observed river discharge with a period of record of thirty years (reference), and simulated river discharges from a hydrological model forced with In-Situ and global precipitation datasets (CHIRPS and MSWEP). The hydrological model is calibrated using independent observed river discharge samples of




five years extracted from the reference time period of thirty years to emulate a data scarce environment, as well as the uncertainty of the available data with a short period of record being fully representative climate variability. The Relative Utility Value of using a particular dataset is determined based on the reference and simulated annual expected utility, which includes the monthly probability of (non-) water scarcity using the irrigation areas obtained and the annual crop production with water

scarcity (not-) happening in a month. The monthly probability of water scarcity will depend on the true (reference observed) water resources availability. Additional production losses are incurred if the irrigation area planned is too large, as then water scarcity conditions will occur more frequently (cost type 2), while too small an area will result in an opportunity cost (cost type 1). The production loss also depends on how sensitive the crop is to water deficit in a particular month. The benefit of using either the In-Situ, CHIRPS or MSWEP datasets in reducing the cost of choosing the irrigation area, irrespective of the

available sample of observed data used in calibrating the model, is evaluated through a Pooled Relative Utility Value, a joined estimate of the Relative Utility Value of the samples of five years.

In the Coello basin in Colombia where the framework was applied, it was found that while the performance metrics of the discharge simulations relate to the Relative Utility Value, the Pooled Relative Utility Value provides a complete hydro-economic indicator to assess the risk of choosing the irrigation area based on observed or simulated discharge data. We find

that for the Coello basin, the CHIRPS precipitation dataset is more beneficial than In-Situ or MSWEP precipitation, as the risk of choosing the irrigation area is lower due to a better estimate of climate variability. For all precipitation datasets evaluated, using a dataset with a length of thirty years leads to a lower risk when compared to using a length of only five years. The risk of choosing the irrigation area based on discharge simulations with thirty years of CHIRPS precipitation is found to be similar to using the observed discharge of five years. Hence, extending the period of record using an extended precipitation dataset to

provide a longer record of discharge simulations (from five to thirty years) compensates the model uncertainty of the model calibration.

In the Coello basin, the global precipitation data CHIRPS is recommended instead of global precipitation data from the MSWEP dataset for estimating surface water availability to support the planning of irrigation areas. This dataset provides a good representation of the climatic variability in this medium sized tropical basin, in part due to the correction of the dataset

using observed station data. While the performance of the available global precipitation datasets would need to be evaluated, the application of the extended hydro-economic framework using global precipitation datasets to force a locally calibrated hydrological model is shown here to support decisions on adequate selection of irrigated areas in Colombia, and can be applied in data scarce basins around the world. Ensuring the use of adequate hydrological information for the estimation of surface water availability through will promote improved decisions for irrigation area planning and prevent economic losses.

**Acknowledgments**

This work received funding from the European Union Seventh Framework Programme (FP7/2007-2013) under grant agreement no. 603608, Global Earth Observation for Integrated Water Resource Assessment (eartH2Observe). We would like to thank IDEAM for providing the discharge and precipitation data.



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





**Table 1. Evaluating the expected annual utility using the planned irrigation area from selected river discharge information relative to expected annual utility using the reference irrigation area.**

| | Using **reference** irrigation area | Using **planned** irrigation area |
|---|---|---|
| Monthly probability of water scarcity | *Annual production under probability of water scarcity* | *Higher or lower annual production under probability of water scarcity* |
| Monthly probability of **no** water scarcity | *Annual production under probability of **no** water scarcity* | *Higher or lower annual production under probability of **no** water scarcity* |
| | Expected annual utility | Higher or lower expected annual utility |





**Table 2. Irrigation areas obtained using different datasets of river discharge information in the Coello basin. The observed river discharge from the complete period of record of 30 years (1983-2012) is the reference information. The irrigation areas are obtained for an agreed water supply reliability of 75% in any one month.**

| Hydrological information used | | Six samples of observed River discharge of 5 years | | | | | |
|---|---|---|---|---|---|---|---|
| | | 1983-1987 | 1988-1992 | 1993-1997 | 1998-2002 | 2003-2007 | 2008-2012 |
| | | Size of the irrigation area (km²) | | | | | |
| Observed River discharge | Q_30 years (reference) | | | 67.45 | | | |
| | Q_5 years | 67.93 | 54.66 | 65.92 | 75.18 | 60.82 | 65.43 |
| | P_In-Situ_30 years | 98.97 | 85.24 | 65.48 | 93.93 | 79.07 | 84.77 |
| | P_In-Situ_5 years * | 51.53 | 47.68 | 57.76 | 79.14 | 42.70 | 52.48 |
| | P_CHIRPS_30 years | 92.38 | 81.72 | 69.37 | 94.56 | 78.32 | 80.97 |
| | P_CHIRPS_5 years * | 47.84 | 40.20 | 61.47 | 84.13 | 43.13 | 45.80 |
| | P_MSWEP_30 years | 105.90 | 99.58 | 86.94 | 113.97 | 94.53 | 98.47 |
| | P_MSWEP_5 years * | 58.56 | 58.40 | 74.17 | 109.09 | 55.61 | 59.69 |





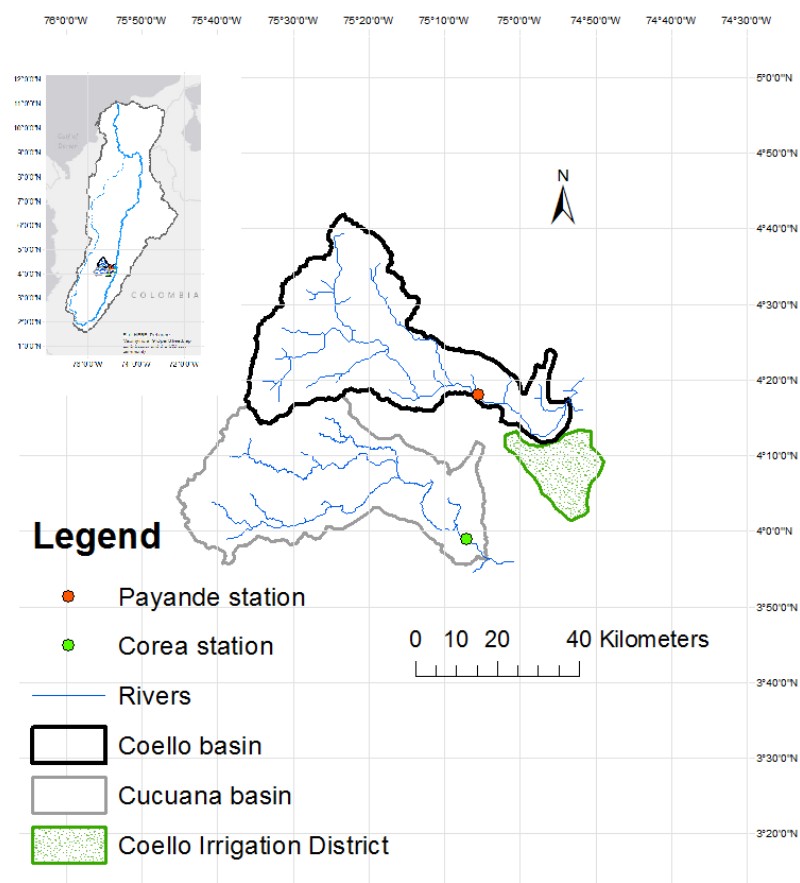

**Figure 1. Map of the Coello and Cucuana River basins and the Coello irrigation district, and their location in the Magdalena macro-basin in Colombia. The points indicate gauging stations.**



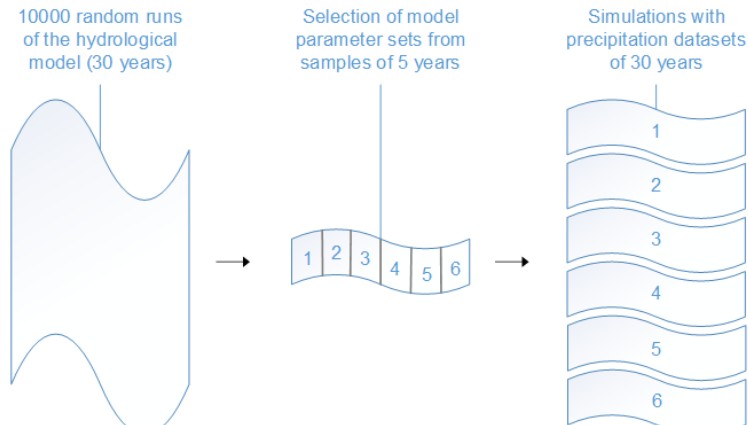

**Figure 2. Obtaining hydrological model simulations from the six samples of 5 years of observed river discharge.**




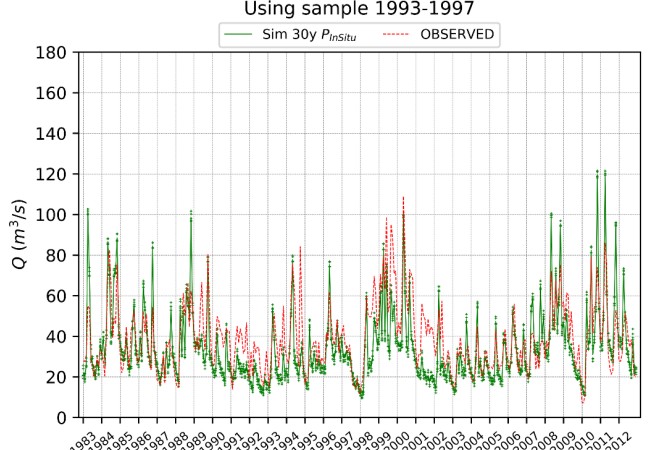
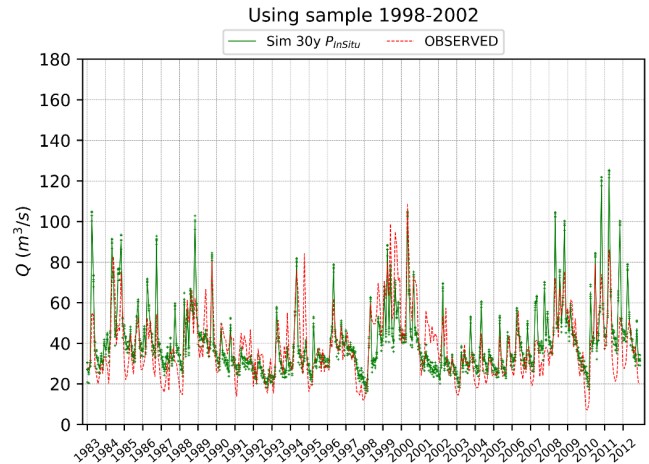

**Figure 3. Observed and simulated discharge for the Coello River at Payende with 30 years (1983-2012) of In-Situ precipitation (Sim 30y $P_{In-Situ}$) for calibration samples 1993-1997 and 1998-2002, with the sample used to calibrate the model indicated in the header.**





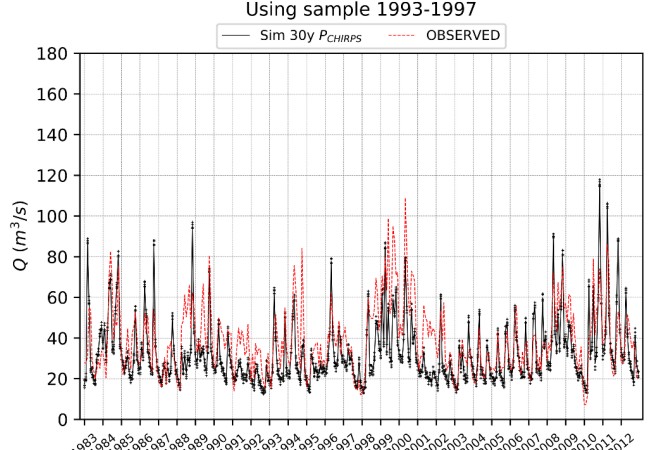
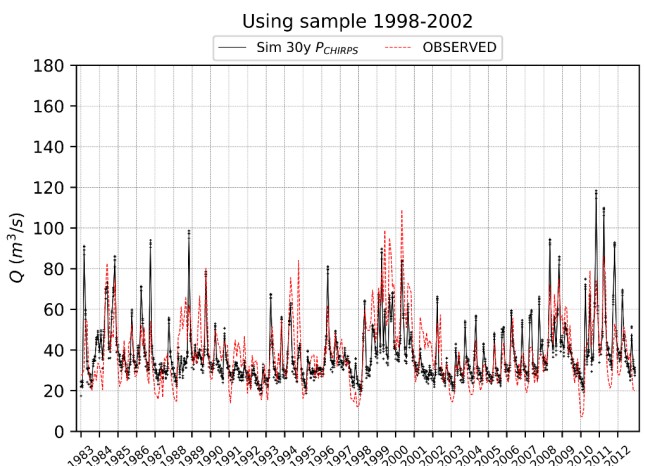

**Figure 4. Observed and simulated discharge for the Coello River at Payende with 30 years (1983-2012) of CHIRPS precipitation (Sim 30y P$_{CHIRPS}$) for calibration samples 1993-1997 and 1998-2002, with the sample used to calibrate the model indicated in the header.**





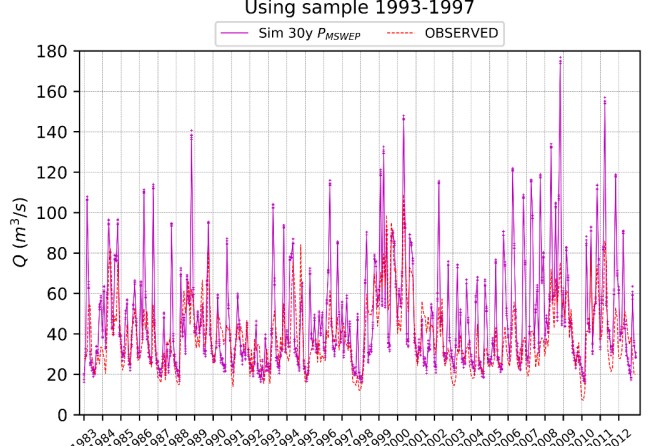
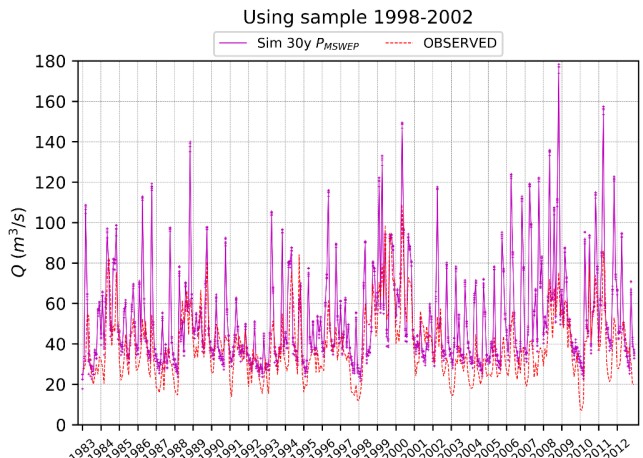

**Figure 5. Observed and simulated discharge for the Coello River at Payende with 30 years (1983-2012) of MSWEP precipitation (Sim 30y P$_{MSWEP}$) for calibration samples 1993-1997 and 1998-2002, with the sample used to calibrate the model indicated in the header.**





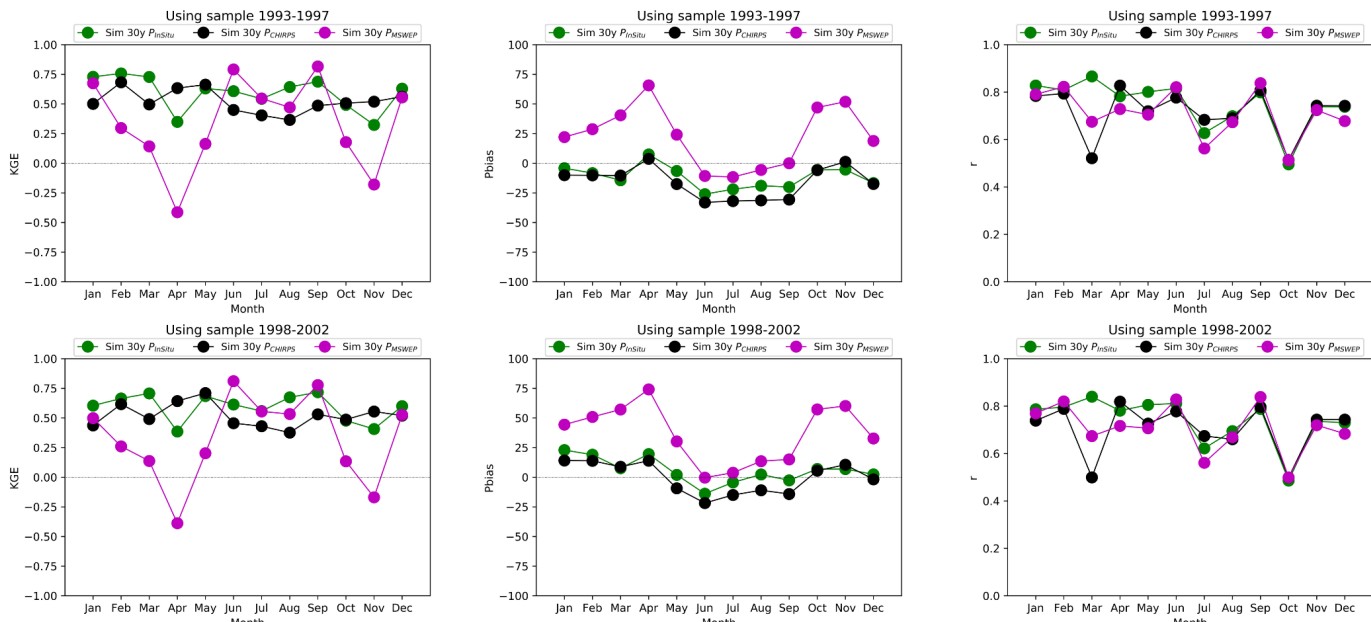

**Figure 6. KGE, Pbias and r performance metric for simulated river discharge for the complete time period of 30 years (1983-2012) using three different precipitation datasets (In-Situ, CHIRPS and MSWEP) in the Coello basin. Two calibration samples are shown (1993-1997, 1998-2002), with the sample used to calibrate the model indicated in the header.**





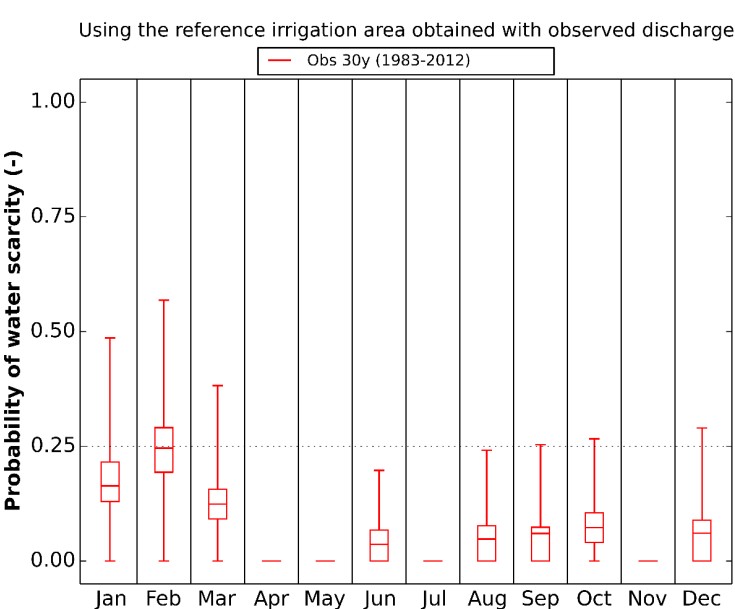

**Figure 7. Probability of water scarcity using the reference irrigation area obtained with the observed river discharge of 30 years (Obs 30y) and the reference surface water availability. Boxplots show the median, interquartile range and minimum-maximum range.**




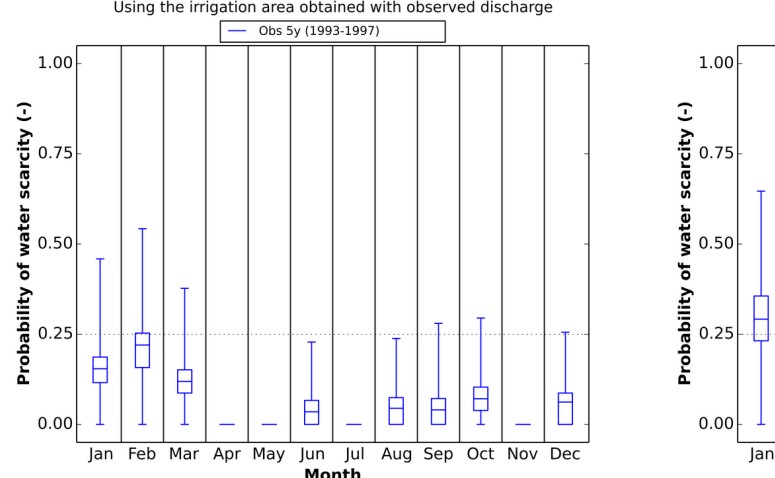
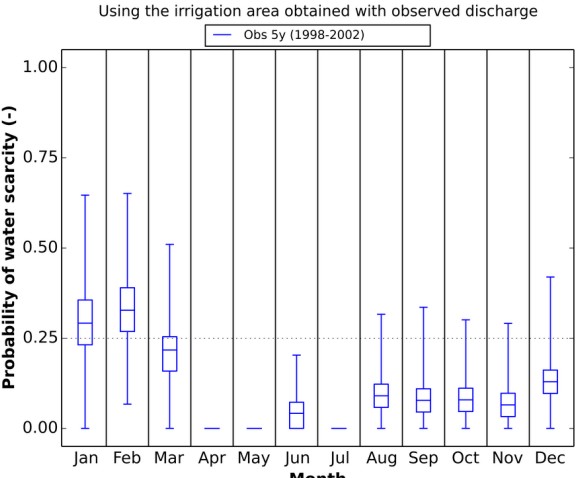

**Figure 8. Probability of water scarcity using the irrigation area obtained with the observed river discharge of 5 years (Obs 5y) and the reference surface water availability. Boxplots show the median, interquartile range and minimum-maximum range.**





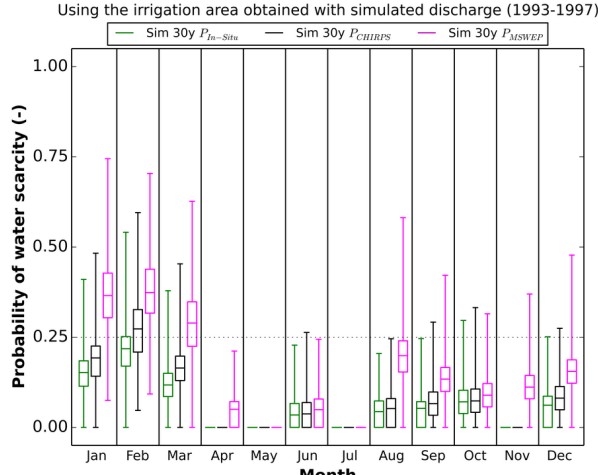
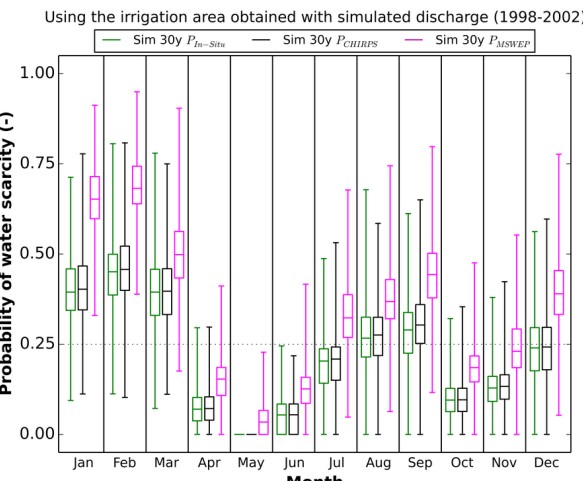

**Figure 9. Probability of water scarcity using the irrigation area obtained with simulated river discharge information (Sim 30y $P_{In-Situ}$, Sim 30y $P_{CHIRPS}$, Sim 30y $P_{MSWEP}$) and the reference surface water availability. Boxplots show the median, interquartile range and minimum-maximum range.**





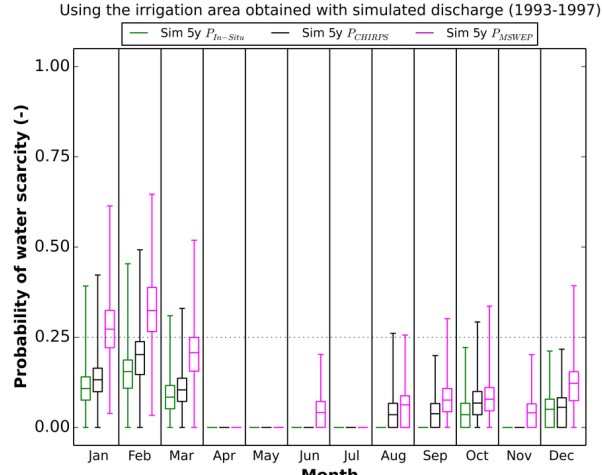
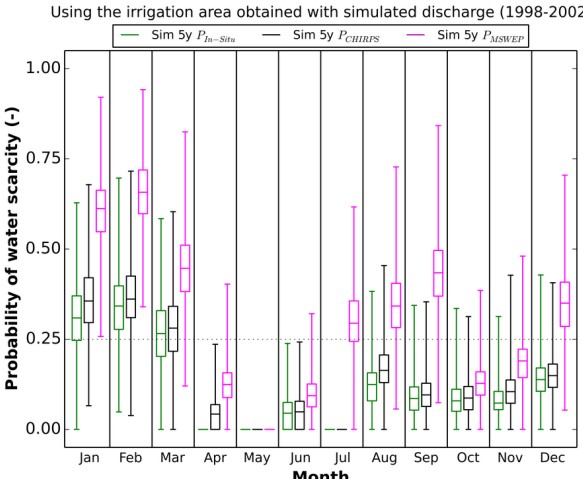

**Figure 10. Probability of water scarcity using the irrigation area obtained with simulated river discharge information (Sim 5y $P_{In-Situ}$, Sim 5y $P_{CHIRPS}$, Sim 5y $P_{MSWEP}$) and the reference surface water availability. Boxplots show the median, interquartile range and minimum-maximum range.**





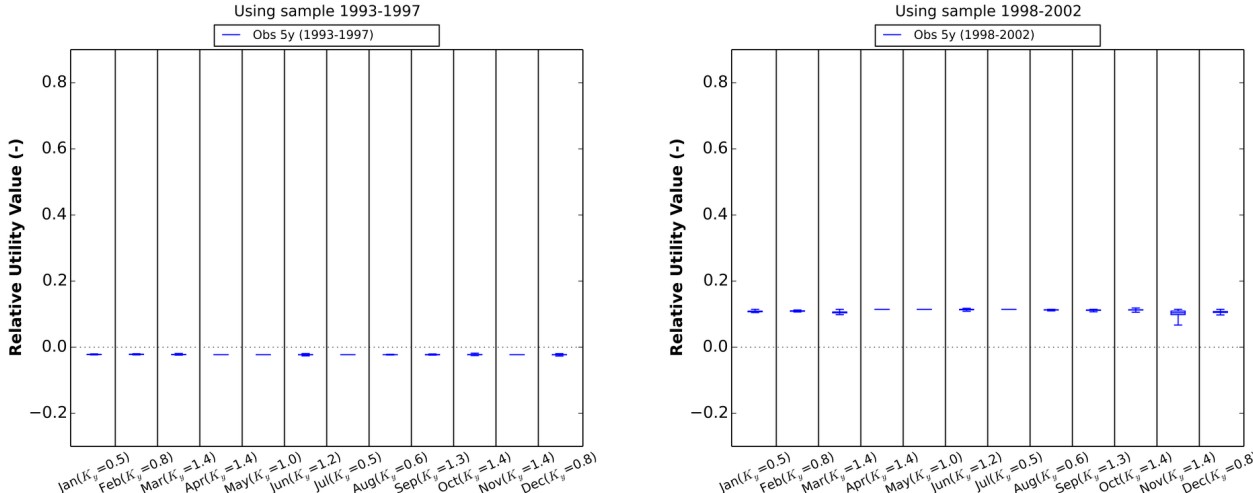

**Figure 11.** Relative Utility Value using observed river discharge of 5 years for water scarcity happening independently in any one month. Ky is the sensitivity of the crop to water deficit. Boxplots show the median, interquartile range and minimum-maximum range.





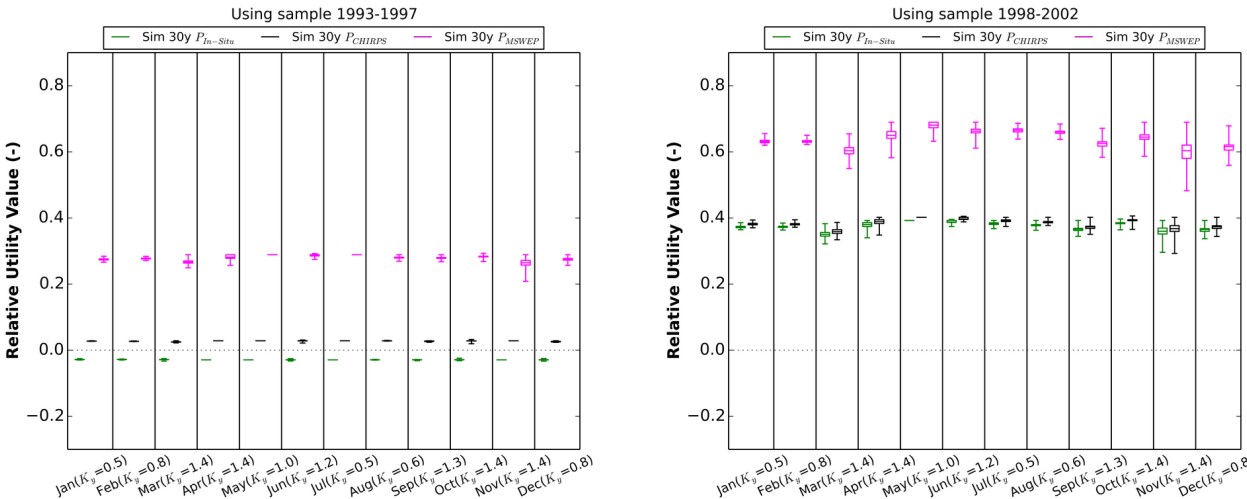

**Figure 12.** Relative Utility Value using simulated river discharge of 30 years for water scarcity happening independently in any one month. Ky is the sensitivity of the crop to water deficit. Boxplots show the median, interquartile range and minimum-maximum range.




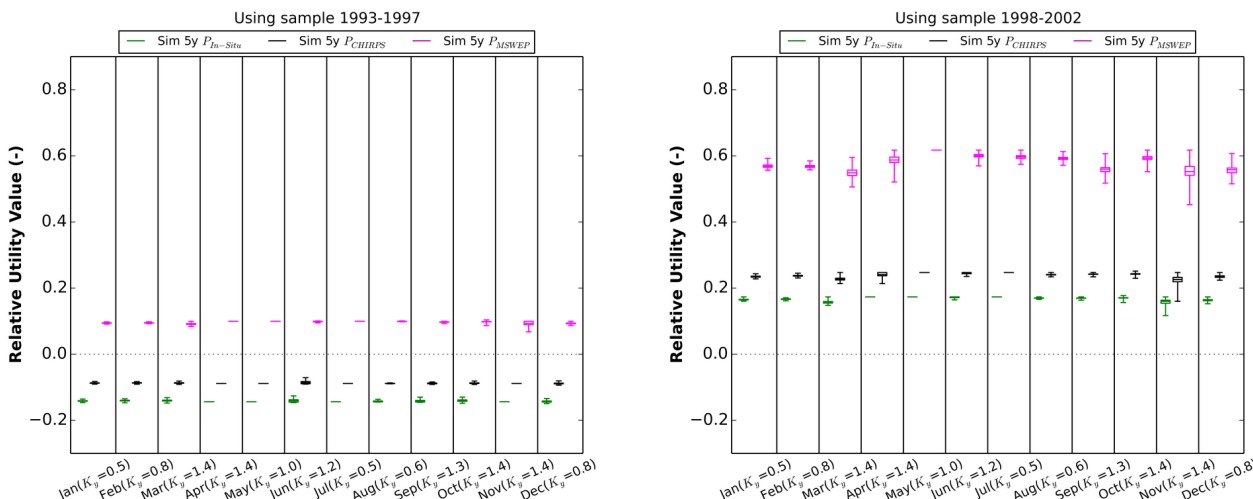

**Figure 13. Relative Utility Value using simulated river discharge of 5 years for water scarcity happening independently in any one month. Ky is the sensitivity of the crop to water deficit. Boxplots show the median, interquartile range and minimum-maximum range.**





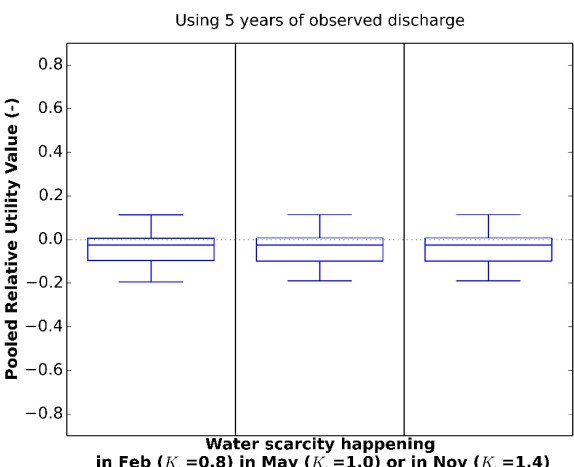

5 **Figure 14. Pooled Relative Utility Value using observed river discharge of 5 years for water scarcity happening independently in February, May or November. Ky is the sensitivity of the crop to water deficit. Boxplots show the median, interquartile range and minimum-maximum range.**





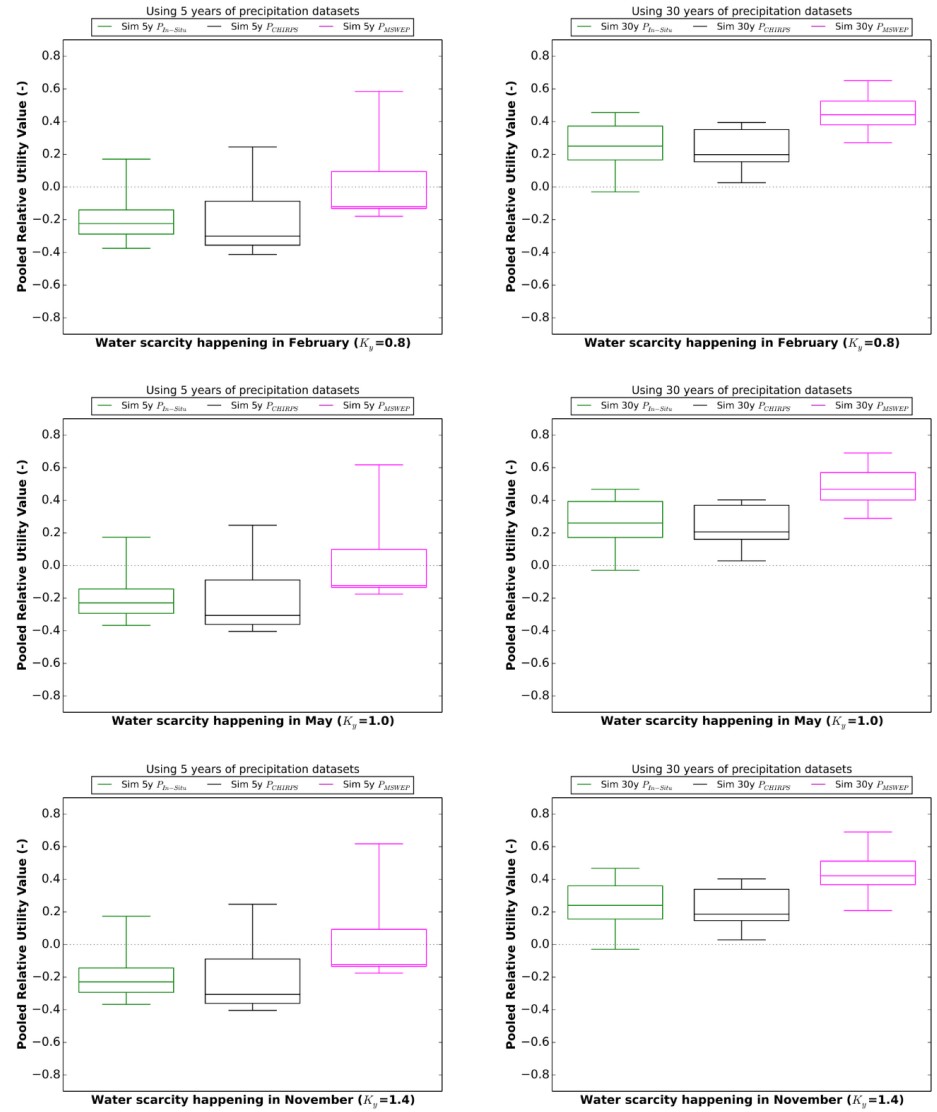

5    **Figure 15. Pooled Relative Utility Value using simulated river discharge of 5 years and 30 years for water scarcity happening independently in February, May or November. Ky is the sensitivity of the crop to water deficit. Boxplots show the median, interquartile range and minimum-maximum range.**

