# Peer review of "Can global precipitation datasets benefit the estimation of the area to be cropped in irrigated agriculture?"

_Hydrology and Earth System Sciences, 2018_

## Referee Comment (RC1) · Anonymous Referee #1 · 9 Nov 2018

**General comments**

The paper aims to demonstrate the added value of global precipitation datasets derived remote sensing data and/or provided by re-analysis for the mapping of areas to be cropped in irrigated areas. This is done in this study by feeding an hydrological model using those global data sets. Surface water availability predicted by the model is then taken as a basis information to estimate the potential cropped areas. The specific question that the paper adresses is : when in situ measurements on water precipitation and water availability are limited in time, can global data sets, compensate their lack of accuracy by the extension of the measurement period thus, allowing to cover more

climatic variability ? The question is particularly relevant. It has already been adressed in several papers from an hydrologic point of view that evaluate indirectly the quality of the global precipitation data sets through an hydrological model and, in particular, the performance of the model to predict streamflow. The main added value of the paper is to go a step further as a second target variable is considered : the potential irrigated area based on water availability. A new metric is introduced based on the work of Kaune et al. (2017) that quantify the risk of sub-optimal water allocation to irrigation. This sub-optimal allocation can lead either to : a production loss if too large irrigation area have been planned or a so-called Âń opportunity cost Âż when too small irrigated area are proposed. The main conclusion of the paper is that the risk of choosing badly planned irrigation area based on discharge simulations with thirty years of one the global precipitation data set (CHIRPS) is found to be similar to using the observed discharge of five years. To my opinion, both the original approach and the results are of great interest for HESS readers.

My greatest concerns are related to the form of the document that I found difficult to follow. The different tools used in the study are described independently in the second part of the manuscript (Âń Method Âż). The result is that the reader discovers the approach and the different steps of the work gradually. The whole approach is (well) summarized in the first part of the discussion section. It could be moved earlier in the text, maybe at the beginning of section 2 and I propose also to add a scheme describing the workflow of the study. In addition, I feel that some important information concerning the hydrologic model description and implementation are lacking (cf. specific comments). Finally, as global precipitation products are at the center of the study and in situ data are available, an evaluation of the global data sets should be conducted as a preliminary step. This could give some quantitative elements for the discussion. Indeed, from the three precipitation data sets compared in this paper (CHIRPS, MSWEP and in situ network data), CHIRPS appear the best suited to plan irrigated areas ... but we don't really know the reasons. Is it because it is able to better catch the real spatio-temporal variability of the catchment precipitation than the in situ data set be-

cause of station scarcity ? Maybe if the worst data set (MSWEP) appears to be strongly biased or strongly inaccurate and that the model calibration is not able to compensate for this data set deficiencies, it could be discarded the rest of the study (after section 3.1 "discharge simulations"). This could significantly simplify the results and discussion sections (and also the figures). Regarding this last point, I believe that the number of figures could be also reduced (cf. specific comments).

Regarding the comments above, I recommend a major revision.

Specific comments

2.1 A better description of the local climate should be provided : at least long term average temperature and precipitation together with average annual evaporative demand. In addition, if flooded rice is the dominating crop in the region. Typical phenological cycle should be given together with the months during which a water stress is critical for production.

2.2 Could you map the network of the meteorological stations (at least provide the number of stations) ? How is measured the discharge ? Is the river bed changing over time?

2.3 The different input of the hydrological model should be described. In particular, how are determined the physical soil and the vegetation characteristics ? Could you please explain how you determine a maximum soil moisture storage capacity max S of 176 mm from the physical soil and vegetation characteristics ? Calibration of the models (L19-29) should be re-written as it is not clear whether the model are calibrated on the full period or on five-years period. I assume the second option has been adopted. The best 5 models from the Monte-Carlo simulation with 10000 ensemble members are then kept which gives five models by 6 periods, 30 calibrated models. I think that the sentence beginning by Âń 10000 models . . . Âż is confusing.

2.4 The rationale of meeting water demand for 75% of the year seems relevant. By

contrast, the demand rate of 0.2 m3/s/km$^2$ is huge as it means 62072 m3/ha/year (common demands for wheat, olive orchard or citrus for instance are between 5000 and 12000 m3/ha/year). Maybe it's ok for flooded rice but a reference should be given. What is the evaporative demand ?

Please better explain how the bootstrap resampling approach is applied to retrieve the empirical distribution of water availability.

(p.4 L15-16) Âń The areas that can be irrigated for each of the six calibrated models Âż Why considering only six models among 30 ? Are only the six best models considered here ?

2.5 Computing a realistic evapotranspiration reduction is not a detail for me as a hydric stress occuring at specific phenological stages of a crop can have dramatic conse-quences on yields and the effects are far to be linear. I agree that a first approximation as the one proposed by the author can be done but please reformulate to explain that a finer description of water stress should be considered in future studies.

3.2 Attributed to model errors and to forcing errors also ?

Âń Finally, for comparison, irrigated areas are derived using only five years of simulated data for each of the six five-year samples, where the simulated five years are the same as the five years used in calibration. Âż Please explain why you choose the five-year sample corresponding to the same used for calibration.

Figures

As already stated, the number of figure could be reduced:

- As the time series of discharge are not clearly readable, please choose one figure between figure 4 and 5.

- Figure 7 and 8 could be merged, it would also ease the comparison between the "real" water scarcity (figure 7 showing water scarcity with the thirty years of observed

river discharge) and two example of five-year periods.

- Likewise, Figure 9 and 10 could also be merged. It would also be better for comparison purposes.

- The RUV using observed river discharges could be superimposed on figure 11 showing the RUV derived from simulated river discharges thus discarding figure 10.

---

## Referee Comment (RC2) · Anonymous Referee #2 · 2 Jan 2019

General Comments:

This paper presents a study on the usefulness of two global precipitation datasets (CHIRPS and MSWEP data) and in-situ data for the estimation of surface water availability for cropped area irrigation planning. A hydrological model forced by those datasets simulates river discharges which are then used to estimate potential irrigated/cropped areas and their relative utility values. The authors show by period-sampling from the available 30 years data the added value of having an extended data records from global datasets. They conclude that this approach permits better calibration of the hydrological model and hence reduces the spread of the so called

pooled relative utility value.

The paper is overall well-structured and represents a significant development effort. Nevertheless, I do see few points in the paper that prevent it from reaching its full potential. I therefore recommend publication of this manuscript with minor revision:

Being the main driver of the study, I would recommend developing more the hydro-meteorological data section with extended description/comparison of the two global precipitation datasets including an proven conclusion on their quality over the study area.

specific comments:

I see a direct link between the hydrological model parameter (evapotranspiration efficiency) and the reduction in evapotranspiration used in the FAO water production function (eq 6) I would recommend the authors to try to establish that link or at least to explain it better.

The authors tend to use long sentences, making it sometime hard to follow, I would recommend rephrasing long sentences into few smaller ones (ex in p3 lines1-3).

Figure S3: colour scheme should be revised

---

## Author Comment (AC1) · 27 Feb 2019

We thank the reviewer for taking the time to review the manuscript and for the helpful comments and suggestions. Here we provide answers to the specific comments and indications of how we propose to improve the manuscript to address the issues raised by the reviewer.

General comments [Referee] The paper aims to demonstrate the added value of global precipitation datasets derived remote sensing data and/or provided by re-analysis for the mapping of areas to be cropped in irrigated areas. This is done in this study by feeding an hydrological model using those global data sets. Surface water availability

predicted by the model is then taken as a basis information to estimate the potential cropped areas. The specific question that the paper addresses is: when in situ measurements on water precipitation and water availability are limited in time, can global data sets, compensate their lack of accuracy by the extension of the measurement period thus, allowing to cover more climatic variability? The question is particularly relevant. It has already been addressed in several papers from an hydrologic point of view that evaluate indirectly the quality of the global precipitation data sets through an hydrological model and, in particular, the performance of the model to predict streamflow. The main added value of the paper is to go a step further as a second target variable is considered: the potential irrigated area based on water availability. A new metric is introduced based on the work of Kaune et al. (2017) that quantify the risk of sub-optimal water allocation to irrigation. This sub-optimal allocation can lead either to: a production loss if too large irrigation area have been planned or a so-called opportunity cost when too small irrigated area are proposed. The main conclusion of the paper is that the risk of choosing badly planned irrigation area based on discharge simulations with thirty years of one the global precipitation data set (CHIRPS) is found to be similar to using the observed discharge of five years. To my opinion, both the original approach and the results are of great interest for HESS readers.

1) My greatest concerns are related to the form of the document that I found difficult to follow. The different tools used in the study are described independently in the second part of the manuscript (Method). The result is that the reader discovers the approach and the different steps of the work gradually. The whole approach is (well) summarized in the first part of the discussion section. It could be moved earlier in the text, maybe at the beginning of section 2 and I propose also to add a scheme describing the workflow of the study.

Reply: We moved the approach of our research that is now described in the discussion section to section 2 (beginning of the method section). Additionally, we added a schematic diagram to clarify workflow of the study (Figure 1). Figure 1. Workflow of

the study to determine the Pooled Relative Utility Value using different irrigation areas obtained from In-Situ, CHIRPS and MSWEP precipitation datasets.

2) In addition, I feel that some important information concerning the hydrologic model description and implementation are lacking (cf. specific comments).

Reply: Indeed, we added a more complete description of the hydrological model and explained the steps taken to implement the model in our research (see reply of specific comments for section 2.3).

3) Finally, as global precipitation products are at the center of the study and in situ data are available, an evaluation of the global data sets should be conducted as a preliminary step. This could give some quantitative elements for the discussion. Indeed, from the three precipitation data sets compared in this paper (CHIRPS, MSWEP and in situ network data), CHIRPS appear the best suited to plan irrigated areas but we don't really know the reasons. Is it because it is able to better catch the real spatio-temporal variability of the catchment precipitation than the in situ data set because of station scarcity? Maybe if the worst data set (MSWEP) appears to be strongly biased or strongly inaccurate and that the model calibration is not able to compensate for this data set deficiencies, it could be discarded the rest of the study (after section 3.1 "discharge simulations"). This could significantly simplify the results and discussion sections (and also the figures). Regarding this last point, I believe that the number of figures could be also reduced (cf. specific comments). Regarding the comments above, I recommend a major revision.

Reply: We added an evaluation of the global precipitation datasets as a preliminary step. We compared global precipitation datasets (CHIRPS and MSWEP) against in situ data in the selected basin. We used performance indicators KGE, percentage of bias (Pbias) and Pearson correlation (r). The evaluation was done for multi-annual monthly precipitation for the selected period 1983-2012 (new Figure 4).

Figure 4. KGE, Pbias and r performance metric for monthly CHIRPS and MSWEP

precipitation in the Coello basin for 30 years (1983-2012).

KGE results show that MSWEP performs better than CHIRPS from October to May. Only in July, MSWEP performs poorly (KGE=-0.1, Pbias=100%). We cannot discard the use of MSWEP neither of CHIRPS. At this stage, we can recommend the use of each dataset for specific months. Important to mention is the fact that CHIRPS and MSWEP are gauge corrected. This would mean that they would both be expected to perform quite well. However, the datasets used to correct each product may differ. That is way we compare the amount of stations used in the Coello basin for each of the precipitation products (In-Situ, CHIRPS and MSWEP). This is helpful for discussing the PRUV results. Even though the amount of stations used is lower for correction in the CHIRPS product (7 stations) compared to the number of stations used in the In-Situ product (14), the results indicate that the satellite information included in CHIRPS still provides a reasonable representation of the basin precipitation. For the MSWEP product the only three stations are used for correction, resulting in a poorer representation of the rainfall in the basin. In summary, the basin precipitation dataset derived from CHIRPS for the Coello basin is better than the MSWEP. The higher resolution of the CHIRPS dataset when compared to that of MSWEP no doubt also contributes in this medium sized, mountainous basin. The poorer comparison of the MSWEP data we found not to be immediately obvious when evaluating the precipitation data using common indicators (e.g. KGE, bias), but was only found when evaluating the hydrological information for determining the irrigated area. As our objective is to evaluate the best hydrological information for irrigation area planning, we do think it relevant to include the RUV and PRUV indicators for all precipitation products, including MSWEP.

Specific comments [Referee] 2.1 A better description of the local climate should be provided: at least long term average temperature and precipitation together with average annual evaporative demand. In addition, if flooded rice is the dominating crop in the region. Typical phenological cycle should be given together with the months during which a water stress is critical for production.

[Figure]

Reply: We provided a better description of the local climate and rice characteristics. The average monthly temperature in the Coello District is 28°C, with maximum daily temperatures reaching 38°C (station 21215080). The reference evapotranspiration is between 137 mm/month in November and 173 mm/month in August with a mean annual evapotranspiration of 1824 mm/year. In the Coello basin the precipitation is bimodal with two peak months in May (186 mm/month) and October (127 mm/month) and two low months in January (50 mm/month) and August (90 mm/month). The mean annual precipitation is 1268 mm/year in the upper basin (station 21215100). Rice total growth length is four months with high sensitivity to water deficit at the flowering stage. The flowering stage starts three months after rice has been planted. Rice is planted throughout the year in different locations inside the irrigation district.

2.2 Could you map the network of the meteorological stations (at least provide the number of stations)? How is measured the discharge? Is the river bed changing overtime?

Reply: We improved the map by including the network of the meteorological stations and providing the number of stations. The map is shown in Figure 2.

Figure 2. Map of the Coello and Cucuana River basins and the Coello irrigation district, and their location in the Magdalena macro-basin in Colombia. The points indicate discharge stations and the squares indicate meteorological stations

The river discharge is measured indirectly with a gauging station managed by the national hydro-meteorological agency in Colombia (IDEAM). IDEAM carries out regular discharge measurements to develop a rating curve. We agree that the accurate estimation of the river discharge from observed water levels using the rating curve will depend on the river bed conditions at the measurement site. However, we did not evaluate the uncertainty of river discharge measurements due to river bed changing over time in this research.

2.3 The different input of the hydrological model should be described. In particular, how

are determined the physical soil and the vegetation characteristics? Could you please explain how you determine a maximum soil moisture storage capacity max S of 176 mm from the physical soil and vegetation characteristics? Calibration of the models (L19-29) should be re-written as it is not clear whether the model are calibrated on the full period or on five-years period. I assume the second option has been adopted. The best 5 models from the Monte-Carlo simulation with 10000 ensemble members are then kept which gives five models by 6 periods, 30 calibrated models. I think that the sentence beginning by "10000 models..." is confusing.

Reply: In section 2.3 we described in more detail how the input data for the hydrological model is generated. We completed the description on how the maximum soil moisture storage capacity ( ) is obtained. An average maximum soil moisture storage capacity of 176 mm was determined for the Coello basin based on the soil texture and the depth of roots in the region. The soil texture and the depth of roots were derived from soil and vegetation maps provided by the Instituto Geográfico Agustín Codazzi in Colombia at a scale of 1:500,000 (IGAC, 2003). Typical values of the available water storage capacity of the soil in millimetres per meters of depth were used based on the soil texture (Shukla, 2013). These values were multiplied by the depth of roots to determine the maximum soil moisture storage capacity in the basin. The calibration of the models (L19-29) will be re-written to make it clear that the model is calibrated for a five year period. Also, Figure 3 was modified to better describe model calibration details. Before the sentence beginning by "10000 models..." we will include the following sentence: "Preliminary a Monte Carlo simulation was developed to obtain the full period of samples and then extracting each sample for calibration".

Figure 3. Obtaining hydrological model simulations from the six samples of 5 years of observed river discharge.

2.4 The rationale of meeting water demand for 75% of the year seems relevant. By contrast, the demand rate of 0.2 m3/s/km is huge as it means 62072 m3/ha/year (common demands for wheat, olive orchard or citrus for instance are between 5000 and 12000

m3/ha/year). Maybe it's ok for flooded rice but a reference should be given. What is the evaporative demand? Please better explain how the bootstrap resampling approach is applied to retrieve the empirical distribution of water availability. (p.4 L15-16) "The areas that can be irrigated for each of the six calibrated models" Why considering only six models among 30? Are only the six best models considered here?

Reply: 75% is a selected target of water supply reliability which is commonly used for large irrigation districts to describe the performance of the system (FAO, 2007; Malano and Hofwegen, 1999; Turral et al., 2010). Of course, this percentage can vary depending on local requirements, rules and policies. In our research, we set this value to compare the irrigation area estimates with different water availability information and the irrigation demand rate. The demand rate used in our study (0.2 $m^3$/s/km) corresponds to the gross irrigation demand rate reported by the local authority (USOCOELLO) in charge of the water management of the Coello irrigation district. This demand rate includes the total efficiency of the irrigation district (application and conveyance efficiencies). We did not determined the total efficiency in the Coello irrigation district, but other studies have established total efficiencies for similar irrigation districts. For this kind of irrigation districts (including rice production, open canal infrastructure and poor maintenance) total efficiencies are typically low, ranging between 30% and 50% (FAO, 2007; Khan et al., 2006). Hence, we can expect high gross irrigation demand values. In addition, the evaporative demand is high. The evaporative demand of the region was provided. Meteorological data and information was included in section 2.1. We extended our description on how the bootstrap resampling approach is applied. Bootstrap resampling is applied for each month for the sample of thirty water availability values (multi-annual monthly values). From this sample we randomly draw X values, and leave these out of the dataset. These are then replaced with X values drawn from the remaining values, thus maintaining the same size of the dataset. This process is repeated 25,000 times. The sentence "The areas that can be irrigated for each of the six calibrated models" means that we want to obtain an irrigation area for each model which was calibrated for each five-year time period. We consider six models of five

years among thirty years. The approach is that we would like to know what happens if we only have five years of available discharge to calibrate a model. To emulate this situation, six independent samples of five years were extracted from the thirty year dataset (1983-1987, 1988-1992, 1993-1997, 1998-2002, 2003-2007 and 2008-2012) for calibration of the model parameters. We are not assuming that these specific six periods lead to the best models. We are assuming that we don't know a priori which period of five years we have available for calibration and how representative the five years that are available of the hydro-climatic variability. Among the six periods we extracted from the full 30 year dataset, we find that some are sampled from more wet periods, while others represent normal or dry periods. Once we select a period of five years the calibration is done, resulting in a model with the "best" model parameter sets, conditional on the sample used for calibration. Similar to the method used in Freer et al. (1996). We changed the sentence in section 2.3. "This resulted in thirty calibrated models" to "This resulted in five calibrated models for six five-year samples". This makes clear that we calibrated the models for each sample and we are using the best five models, conditional on the sample used for calibration.

2.5 Computing a realistic evapotranspiration reduction is not a detail for me as a hydric stress occuring at specific phenological stages of a crop can have dramatic consequences on yields and the effects are far to be linear. I agree that a first approximation as the one proposed by the author can be done but please reformulate to explain that a finer description of water stress should be considered in future studies.

Reply: We agree that we have simplified the approach in obtaining the reduction in evapotranspiration for determining the crop yield reduction. We selected 20% in evapotranspiration reduction for the reference irrigation area when water scarcity occurs. We selected this value because rice farmers can easily cope with such an evapotranspiration reduction as corresponding yield reduction values are reasonable (FAO, 2012). However, we did include the crop response factor Ky for each crop phenological stage (also called crop development stage), which allowed us to evaluate the rice

yield reduction at specific development stages. Hence, our yield reduction results will depend on which month water scarcity occurs. In our experiment each month corresponds to a crop development stage. For example, the total growth length for rice is four months with the highest sensitivity to water deficit at the flowering stage (at the third month after being planted). This means that when water scarcity occurs at the flowering stage the rice yield reduction is higher compared to the rice yield reduction when water scarcity occurs at any other crop development stage.

3.2 Attributed to model errors and to forcing errors also? "Finally, for comparison, irrigated areas are derived using only five years of simulated data for each of the six five-year samples, where the simulated five years are the same as the five years used in calibration" Please explain why you choose the five-year sample corresponding to the same used for calibration.

Reply: We choose the five year sample corresponding to the same used for calibration. The approach was used to evaluate the contribution of model uncertainty in the estimates of irrigated area. When using the same five years of simulate river discharge information (including the spread due to model uncertainty), and compare the reference areas established using the observed river discharge of five years, then the difference can be attributed to model uncertainty. This we then used to compare irrigated area estimates using five years against thirty years of simulated discharge information. In this way, we can explicitly determine the benefit of using the additional time period (30 years against 5 years) of river discharge information for planning irrigated areas.

Figures As already stated, the number of figure could be reduced: - As the time series of discharge are not clearly readable, please choose one figure between figure 4 and 5.

Reply: Figure 4 was chosen for CHIRPS (now Figure 5).

- Figure 7 and 8 could be merged, it would also ease the comparison between the "real" water scarcity (figure 7 showing water scarcity with the thirty years of observe

river discharge) and two example of five-year periods.

Reply: Figure 7 and 8 have been merged (now Figure 7).

- Likewise, Figure 9 and 10 could also be merged. It would also be better for comparison purposes

Reply: Figure 9 and Figure 10 have been merged. (now Figure 8).

- The RUV using observed river discharges could be superimposed on figure 11 showing the RUV derived from simulated river discharges thus discarding figure 10.

Reply: We did not superimpose observed river discharges on Figure 11 showing the RUV derived from simulated river discharges. This is because we think it is clearer to show only RUV, Obs 5y results in Figure 11 (now Figure 9) because it is then consistent with Figure 14 (now Figure 11) showing only PRUV, Obs 5y results. However, we did merge Figure 12 and Figure 13, resulting in Figure 10 which now shows the Sim 5y and Sim 30y (similar as we did for the probability of water scarcity in Figure 8).

Please also note the supplement to this comment:
https://www.hydrol-earth-syst-sci-discuss.net/hess-2018-331/hess-2018-331-AC1-supplement.pdf

———————————————

```
                              ┌─────────┐
                              │  Start  │
                              └─────────┘
                                   │
   Data collection: River discharge, precipitation, temperature, root soil depth and soil texture of the basin
```

Start

Data collection: River discharge, precipitation, temperature, root soil depth and soil texture of the basin

Observed surface water availability 30 years

In-Situ ETP 30 years

In-Situ precipitation 30 years

Hydrological modelling using 10000 Monte Carlo runs

Samples of 5 years

Selection of model parameter sets

MSWEP precipitation 30 years

Simulation of surface water availability for 30 years and for 5 years (each sample)

CHIRPS precipitation 30 years

Reliability of water supply to satisfy irrigation demand

No

Reliability is higher than target reliability?

Yes

Irrigation area

Probability of water scarcity for each irrigation area

Crop yield

Relative Utility Value

Yield reduction factor

Pooled Relative Utility Value

End

**Fig. 1.**

[Figure]

**Fig. 2.**

[Figure]

**Fig. 3.**

Hydrological modelling using 10000 Monte Carlo runs (30 years)

Selection of model parameter sets from samples of 5 years

Simulation of surface water availability for 30 years and for 5 years

1 2 3 4 5 6

2
3
4
5

**Fig. 4.**

---

## Author Comment (AC2) · 27 Feb 2019

We thank the reviewer for taking the time to review the manuscript and for the helpful comments and suggestions. Here we provide answers to the specific comments and indications of how propose to improve the manuscript to address the issues raised by the reviewer.

General comments [Referee] This paper presents a study on the usefulness of two global precipitation datasets (CHIRPS and MSWEP data) and in-situ data for the estimation of surface water availability for cropped area irrigation planning. A hydrological model forced by those datasets simulates river discharges which are then used to estimate potential irrigated/cropped areas and their relative utility values. The authors show by period-sampling from the available 30 years data the added value of having an extended data records from global datasets. They conclude that this approach permits better calibration of the hydrological model and hence reduces the spread of the so called pooled relative utility value. The paper is overall well-structured and represents a significant development effort. Nevertheless, I do see few points in the paper that prevent it from reaching its full potential. I therefore recommend publication of this manuscript with minor revision: • Being the main driver of the study, I would recommend developing more the hydro-meteorological data section with extended description/comparison of the two global precipitation datasets including a proven conclusion on their quality over the study area.

Reply: This was also raised by the first reviewer and we developed a more in depth hydro-meteorological data section with extended description and comparison and quality evaluation of the global precipitation datasets. We added an evaluation of the global precipitation datasets as a preliminary step. We compared global precipitation datasets (CHIRPS and MSWEP) against in situ data in the selected basin. We used performance indicators KGE, percentage of bias (Pbias) and Pearson correlation (r). The evaluation was done for multi-annual monthly precipitation for the selected period 1983-2012 (new Figure 4).

Figure 4. KGE, Pbias and r performance metric for monthly CHIRPS and MSWEP precipitation in the Coello basin for 30 years (1983-2012).

KGE results show that MSWEP performs better than CHIRPS from October to May. Only in July, MSWEP performs poorly (KGE=-0.1, Pbias=100%). We cannot discard the use of MSWEP neither of CHIRPS. At this stage, we can recommend the use of each dataset for specific months.

Specific comments [Referee] I see a direct link between the hydrological model parameter (evapotranspiration efficiency) and the reduction in evapotranspiration used in the

[Figure]

FAO water production function (eq 6) I would recommend the authors to try to establish that link or at least to explain it better.

Reply: We agree that these could be considered linked concepts. However, the hydrological model parameter is a parameter used to simulate the surface water availability in the basin. This is used to derive the water availability to the district for irrigation. The reduction in evapotranspiration is used to characterise the crop yield response due to water shortage in the irrigation district. We will add a sentence to the manuscript to comment on the conceptual link, but also underline the difference.

The authors tend to use long sentences, making it sometime hard to follow, I would recommend rephrasing long sentences into few smaller ones (ex in p3 lines1-3).

Reply: We will improve the readability by reducing the length of the sentences and rephrasing where possible/appropriate.

Figure S3: colour scheme should be revised

Reply: We will revise the colour scheme in Figure S3.

Please also note the supplement to this comment:
https://www.hydrol-earth-syst-sci-discuss.net/hess-2018-331/hess-2018-331-AC2-supplement.pdf

———————————————————

[Figure]

**Fig. 1.**